Systemic acquired resistance inducing chemicals mitigate black scurf disease in potato by activating defense-related enzymes

Buswal Manoj Kumar 1
Punia Rakesh 1
Kumar Mukesh 1
Tiwari Rahul Kumar 2
Lal Milan Kumar milan2925@gmail.com 2
Kumar Ravinder chauhanravinder97@gmail.com 2 3
1 Plant Protection, Chaudhary Charan Singh Haryana Agricultural University , Hisar , Haryana , India
2 ICAR-Central Potato Research Institute , Shimla , Himachal Pradesh , India
3 ICAR-Indian Agricultural Research Institte , Delhi , New Delhi , India
Yin Heng
Electronic publication date: 2024 Nov 18
Publication date: 2024
Volume: 12
Electronic Location ID: e18470
Received 2024 Feb 21; Accepted 2024 Oct 15
Copyright: ©2024 Buswal et al.
Copyright year: 2024
Copyright holder: Buswal et al.
License: This is an open access article distributed under the terms of the Creative Commons Attribution License, which permits unrestricted use, distribution, reproduction and adaptation in any medium and for any purpose provided that it is properly attributed. For attribution, the original author(s), title, publication source (PeerJ) and either DOI or URL of the article must be cited.
License URL: https://creativecommons.org/licenses/by/4.0/

Keywords: Induced resistance, Antioxidants, Defense, Signaling, Potato, Rhizoctonia

Funding: The authors received no funding for this work.

==============================
The potato, being an underground vegetable crop, faces consistent threats from soil- and tuber-borne fungal and bacterial pathogens. Black scurf and stem canker disease caused by the fungal pathogen Rhizoctonia solani Kuhn is a critical global concern in the potato cultivation system. In this study, we evaluated the disease mitigation potential of five systemic acquired resistance-inducing chemicals viz., salicylic acid, jasmonic acid, β-aminobutyric acid, γ-aminobutyric acid and hydrogen peroxide (H2O2). Two common methods, tuber dipping and foliar spray, were utilized in this experiment to evaluate pathogen inhibition on inoculated tubers. The results revealed that all the systemic acquired resistance inducing chemicals were effective in disease suppression in a concentration-dependent manner compared to an inoculated control. Significant differences (P < 0.005) were evident among the various treatment combinations, with salicylic acid being the most effective in alleviating black scurf disease. Maximum reduction in disease incidence compared to the control was observed with salicylic acid (57.89% and 73.68%), followed by jasmonic acid (52.63% and 65.78%) and H2O2 (49.99% and 60.52%) under the tuber dipping treatment combinations. Whereas, in the foliar application, the maximum reduction in disease incidence compared to the control was observed with salicylic acid (44.73 and 63.15%), followed by jasmonic acid (42.10 and 60.52%) and H2O2 (39.46 and 52.63%). The tuber dipping treatments were significantly more efficacious (P < 0.005) compared to foliar spray for all treatment combinations. The biochemical analysis of defense-related enzymes and metabolites demonstrated the induced resistance activation under these treatments. The activity of peroxidase, polyphenol oxidase, and phenyl ammonia-lyase was significantly higher in treated tubers as compared to inoculated and uninoculated control. The total phenol content was also elevated in treated tubers as compared to the respective control. Altogether, these resistance-inducing chemicals can be successfully included in integrated disease management programs.

Introduction

Potato belonging to the genus Solanum of family Solanaceae is cross-pollinated herbaceous annual crop having tetraploid genome organization (Kumar et al., 2021b; Lal et al., 2022). The crop has worldwide importance due to its diverse distribution, high dry matter production per unit of land, nutritional contribution, and value as a source of income and employment in developing nations (Campos & Ortiz, 2019). Raw potato contains 79 per cent water, 17 per cent carbohydrates (88% of which is starch), 2 per cent protein, negligible fat and provides 77 kcal/100 g along with a good amount of vitamin B6 and vitamin C (Camire, 2016; Devaux et al., 2019). Consumption wise potato hold the third position as staple food in world crop after wheat and rice (FAOSTAT 2020). The major portion of potato harvest (two-third of total produce) is consumed as fresh staple food by around 1.3 billion people.

Potato production is severely affected by biotic and abiotic stressors (Tiwari et al., 2020; Tiwari et al., 2021b). Additionally, potatoes grown using traditional vegetative growth techniques are frequently vulnerable to infections from bacteria, viruses, and fungi, leading to poor quality and reduced production potential (Kumar et al., 2021b). The major diseases affecting potato crop worldwide are late blight (Phytophthora infestans), early blight (Alternaria solani), black scurf (Rhizoctonia solani), brown rot or bacterial wilt (Ralstonia salanacearum), black leg and soft rot (Erwinia carotovora and E. atroseptica). Additionally, viruses such as Potato leaf roll virus, Potato virus X, Potato virus Y and Geminivirus (Tomato leaf curl new delhi virus) are a consistent threat to potato cultivation (Díaz Arias, Munkvold & Leandro, 2011; Tiwari et al., 2021a). Amongst these crucial diseases, black scurf disease has appeared as a major problem in recent years (Larkin & Brewer, 2020).

The polyphagous fungal causal agent Rhizoctonia solani  Kuhn (teleomorph: Thanatephorus cucumeris (Frank) Donk) is mainly responsible for the occurrence of black scurf and stem canker disease in potato (Selva Kumar et al., 2013). The soil-borne necrotrophic pathogen, which affects a variety of field and horticultural crops, is widespread throughout all potato-growing regions of the world. Potato stems and stolons are adversaly affected by the Rhizoctonia solani fungus on plantlets. In addition to irregular brown to black hard masses of sclerotia, which are the classic symptoms of black scurf disease in potato tubers, the original mother tuber infection also causes poor crop stands, impaired plant growth, reduced tuber number and size, and distorted tubers (Arora & Khurana, 2006). The prevalence of the pathogen not only limits the crop’s productivity but also causes quantitative and qualitative damage to the potato crop. Early-season infection of the stem, stolon, and primary roots causes quantitative loss because it suppresses the sprouts, which ultimately leads to a smaller tuber. Late in the growing season, however, the surfaces of the progeny tubers are covered with hard sclerotia masses, leading to deformities. The development of progeny tuber-borne sclerotia degrades the quality of the tubers and lowers their market value (Jager et al., 1991). Previously, a marketable yield loss of 15% in hilly regions and 10% in plains was also reported for major potato-growing zones in India (Lal et al., 2017). An estimated yield loss of 30% has been reported in Canada.

Fungicide chemicals are currently necessary for the control of black scurf in potatoes. Fungicides such benomyl, carbendazim, thiabendazole, pencycuron, thiophinate methyl, and pyaclostrobin are applied to seeds to suppress the disease (Lal et al., 2017). The two compounds that are most frequently utilised in Indian conditions to treat black scurf are boric acid and pentacyron (Paul Khurana, 2006; Arora & Khurana, 2006). Along with chemical-based management practices some eco-friendly methods have also been employed for an effective management of this disease. Various organic amendments, including vermicompost, Neem cake, and farmyard manure (FYM), were found to effectively suppress the growth of the disease in affected tubers. Vermicompost and neem cake were documented to have restricted the pathogen growth compared to the control (Rahul et al., 2014). Additionally, it was discovered that composted cattle manure diminished the propagule density of R. solani in soil (Kuter et al., 1983), mostly because antagonistic bacteria such Trichoderma species, Gliocladium genera, Pseudomonas fluorescens, and Bacillus species were more active. It has been revealed that diverse isolates of Trichoderma spp. are capable of suppressing tuber-borne pathogenic fungi such Sclerotium rolfsii, Pythium species, Fusarium sambucinum and R. solani (Bernard et al., 2012).

Induced resistance is a mechanism that allows plants to increase their degree of baseline defense against pathogen attack. It can be induced by a variety of abiotic and biotic stimuli (Khan et al., 2020). Systemic acquired resistance (SAR) and induced systemic resistance (ISR), which vary depending on the type of elicitor and the regulatory mechanisms involved, are two classic examples of inducible plant defence (Tiwari et al., 2021b; Naga et al., 2021). Jasmonic acid (JA), ethylene, and the analogue of salicylic acid (SA), acibenzolar-S-methyl (ASM), are known to participate in signalling pathways that induce host resistance (Kumar et al., 2021a; Lal et al., 2021). The quantities of phytoalexins, lignin, and phenolic compounds, as well as the activity of chitinase, phenylalanine ammonia-lyase, and peroxidase, indeed play a significant role in strengthening host resistance to infections (Bagy et al., 2019). The use of inducers of resistance has emerged as a substitute to alleviate the yield losses forced on by this disease due to the low efficacy of fungicides and the lack of potato cultivars with adequate levels of resistance to black scurf. Therefore, the present study was formulated with the major objective of exploration of environment-friendly management options for black scurf management in potato. The various levels of induced resistance coupled with associated defense-related enzyme activities were studied in treated plants to establish the correlation of spray or tuber dip-based effective management of disease using sustainable phytoprotectants.

Material and Methods

Inoculation of potato tubers under the screen house conditions

Sterilized field soil was placed in plastic pots, and at a depth of 5.0 cm, 20 g of R. solani inoculum per pot was added. For the assessment, a susceptible variety of potato named “Kufri Bahar” was used. Three replications of each treatment were kept in the experiment. Three weeks prior to planting potato tubers under screen house conditions, sterilized soil combined with various organic amendments at 10 and 20 g/kg soil was placed in new plastic pots. This was done to test the effectiveness of the various organic amendments. The inoculum was prepared through pure culture of Rhizoctonia solani mass multiplication on PDA and PDB followed by incorporation the pots. A different set of plastic pots with sterilized soil also maintained for the efficacy of systemic acquired resistance (SAR) inducing chemicals (Table 1) against R. solani at two concentrations i.e., at 150 and 250 µg/ml for all chemical inducers separately by tuber dipping for 10–15 min and foliar sprays at tuber initiation stage i.e., after 40 days of planting. Pure water treatments were also maintained as inoculated control and un-inoculated control for SAR inducing chemicals.

Table 1 Treatments of different systemic acquired resistance (SAR) inducing chemicals against R. solani under screen house conditions.

Treatment no.	Treatment	
T1	Salicylic acid (SA)	
T2	Jasmonic acid (JA)	
T3	β-aminobutyric acid (BABA)	
T4	γ-aminobutyric acid (GABA)	
T5	Hydrogenperoxide (H2O2)	
T6	Control	
T7	Un-inoculated control (with water)	

Assessment of disease development under screen house conditions

The efficacy of systemic acquired resistance (SAR) inducing chemicals against R. solani was observed at 150 and 250 µg/ml by tuber dipping and foliar sprays at tuber initiation stage in different set of plastic pots with sterilized soil maintained under screen house conditions. Un-inoculated control was also maintained for SAR inducing chemicals. The data of disease incidence was recorded as per previous reports of Lal et al. (2017).

Biochemical analysis of the plants after application of SAR chemicals

Leaves from the treated potato tubers and the control plants were taken at different intervals (30, 45 and 60 days), whereas, the leaves from the spray treated potato plant and control plants were taken at 0, 1, 3 and 6 days from screen house. The collected leaves of each sample were immediately homogenized on ice to retain enzymatic activity. These samples were powdered and further used to study for diverse chemical alterations might have occured in the host tissue.

Estimation of total phenols

The total phenols content was estimated as per the procedure adopted by Zieslin & Ben-Zaken (1993). The reagents mainly used in this experiment consisted of 80% ethanol, 1N Folin Ciocalteau’s phenol reagent, totally saturated Na2CO3 solution prepared by dissolving 17.5 g Na2CO3in 50 ml of distilled water, standard catechol solution (10 mg of catechol/100 ml of distilled water). Fresh potato leaves weighing 1 g were weighed and pulverised in 10 ml of 80% ethanol using a pestle and mortar. The centrifugation was performed on homogenate at 10,000 rpm for a duration of 30 min. The supernatant collection and dehydration experiment was performed subsequently. The residue was then dissolved in 5 ml of distilled water, to which 0.2 ml of the extract and 0.25 ml of the Folin-Ciocalteau reagent (1N) were added. After three minutes, one ml of 20% Na2CO3was added, and the mixture was heated in a water bath for one minute before cooling. At 725 nm, the absorbance was calculated against a blank for the reagent. An appropriate standard curve was drawn using different concentrations (µg/ml) of catechol and from the origin a straight line was drawn. The point lying on or near the straight line was used to calculate ‘x’.

Concentration of the point

x = ————————————————————–

Absorbance of the point

x × sample absorbance = y (µg/ml)

y × 15 (dilution factor) = z (µg/g)

z/1000 = z′ (where z′ was total phenols in terms of µg/g)

Total phenols = z′ × 100 (mg/100 g FW).

Thus the total quantity of phenols in the potato leave samples was calculated and expressed in terms of µg catechol g−1FW.

Preparation of enzyme extract

Enzymatic experiments on peroxidase and polyphenol oxidase were conducted using potato leaves. In a cold sampling bag, leaves from the potato plant were collected. With the aid of filter paper, the leaves were gently wiped to remove any excess water from the surface after being thoroughly rinsed with cold, distilled water. To ensure maximum enzyme extraction from the leaves in the extract, standardised extraction conditions with respect to buffer molarity and pH were maintained. To get the most enzymatic activity, extraction was done between 0 and 4 degrees Celsius. The leaf sample was ground in a cold mortar and pestle with 4 ml of a pH-neutral, 0.1 M potassium phosphate buffer (7.0). In a chilled centrifuge equipment, the homogenate was spun at 12,000 rpm for half an hour. The supernatant was carefully separated and used for the enzyme assay, to determine the activity of enzymes such as peroxidase (POX) and polyphenol oxidase (PPO).

Assay for peroxidase

The peroxidase activity was estimated as per the previous report (Shannon, Kay & Lew, 1968). The primary ingredients included 0.05 M potassium phosphate buffer (pH 6.5), 0.1% ortho-dianisidine solution in methanol, and 6.0% H2O2 produced using 0.1 M potassium phosphate buffer (pH 6.5). The mixing of 0.5 ml of O-dianisidine solution, 3.5 ml of 0.05 M potassium phosphate buffer (pH 6.5), and 100 µL of extract was performed. After transferring the contents to the spectrophotometer’s cuvette, 0.1 ml of H2O2 was added to initiate the reaction. The reaction mixture without H2O2 served as a control. An increase in absorbance at 430 nm was observed at 30-second intervals over three minutes. The units of change in absorbance min−1 g−1 fresh weight were used to express enzyme activity.

Assay for polyphenol oxidase

The method of Zauberman et al. (1991) was followed to estimate polyphenol oxidase. The experiment’s reagents included 0.05 M potassium phosphate buffer (pH 6.5) and 0.1 M catechol. Following the addition of 0.5 ml of 0.1 M catechol, 1 ml of 0.05 M potassium phosphate buffer (pH 6.5) was added to 0.5 ml of enzyme. For three minutes, the absorbance was measured at 410 nm at 30-second intervals. The units of change in absorbance min−1 g−1 fresh weight used to express enzyme activity on fresh weight basis.

Assay for phenylalanine ammonia lyase

The phenylalanine ammonia lyase was estimated utilising the approach employed by Dickerson et al. (1984). 500 mg of leaves were homogenised in 5 ml of cold, 25 mM borate HCl buffer (pH 8.8) with 5 mM -mercaptoethanol. The supernatant from the homogenate was used as an enzyme source after being centrifuged for 15 min at 4 °C and 15,000 rpm. The test combination consisted of 1.3 ml of water, 0.5 ml of borate buffer, and 0.5 ml of enzyme extract. 1 ml of 12 mM L-phenylalanine was added to begin the procedure. The reaction mixture was incubated for one hour at 32 degrees Celsius. The reaction was halted by the addition of 0.5 cc of 2N HCl. By combining phenylalanine with 2N HCl, phenylalanine was used as a control. At 290 nm, the absorbance was measured. A standard curve made from t-cinnamic acid was used to quantify the concentration of the compound. In terms of fresh weight, the enzyme activity was given as mol trans-cinnamic acid min−1.

Statistical analysis

The data from all the lab and field experimental observations were analyzed using the statistical package of the OPSTAT program (HAU, Hisar, Haryana, India). The data is represented in the form of means of three replications, followed by statistical analysis with standard error (SE) and critical difference (CD) at p = 0.05. Post-hoc comparisons were performed to determine the differences between treatments, concentrations, and their interactions.

Experimental Results

Effect of SAR-inducing chemicals on black scurf disease suppression of potato under screen house conditions

The efficacy of systemic acquired resistance (SAR)-inducing chemicals against R. solani was observed at 150 and 250 µg/mL concentrations. This evaluation was conducted through tuber dipping and foliar sprays during the tuber initiation stage in different sets of plastic pots filled with sterilized soil, all maintained under screen house conditions (Figs. 1A and 1B).

Figure 1 (A) Effect of tuber dip treatments on disease incidence in artificially inoculated tubers (B) effect of foliar spray on disease incidence of plants challenge inoculated with inoculated pathogen.

Bars are represented as mean values ± SEM of three replicates. *** and * values in the figure represent significant (P < 0.05) differences while “ ns” denotes non significant difference. Abbreviations: BABA, β-Aminobutyric acid; GABA, γ-Aminobutyric acid and H2O2, Hydrogen peroxide.

Tuber dipping

The results presented in the Fig. 1A revealed minimum disease incidence with salicylic acid (26.67 and 16.67%) followed by jasmonic acid (30.00 and 21.67%), hydrogen peroxide (31.67 and 25.00%), BABA (38.34 and 33.34%) and GABA (41.67 and 35.00%) as compared to inoculated control (63.33%) and un-inoculated control (06.70%) at both the concentrations i.e., 150 and 250 µg/ml, respectively. All the SAR activators were found significantly superior in reducing disease incidence over the control. Maximum reduction in disease incidence over control was observed with SA (57.89 and 73.68%) followed by JA (52.63 and 65.78%) and H2O2 (49.99 and 60.52%) over control at both the concentrations i.e., 150 and 250 µg/ml, respectively. Minimum reduction in disease incidence was observed with GABA (34.20 and 44.73%) followed by BABA (39.46 and 47.36%).

Foliar spray

The data presented in Fig. 1 (B) revealed that minimum disease incidence was recorded with salicylic acid (35.00 and 23.34%) followed by jasmonic acid (36.67 and 25.00%), hydrogen peroxide (38.34 and 30.00%), BABA (43.34 and 40.00%) and GABA (45.00 and 41.67%) as compared to inoculated control (63.33%) and un-inoculated control (06.70%) at both the concentrations i.e., 150 and 250 µg/ml, respectively. All the SAR activators were found significantly superior in reducing disease incidence over the control. Maximum reduction in disease incidence over control was observed with SA (44.73 and 63.15%) followed by JA (42.10 and 60.52%), H2O2 (39.46 and 52.63%) and BABA (31.56 and 36.84%) over the control at both the concentrations i.e., 150 and 250 µg/ml, respectively.

Biochemical analysis

Biochemical changes occurring in the host following application of systemic acquired resistance-inducing chemicals

The efficacy of systemic acquired resistance (SAR) inducing chemicals was observed at 150 and 250 µg/ml against R. solani by tuber dipping and foliar sprays at tuber initiation stage in different set of plastic pots with sterilized soil maintained under screen house conditions. After the application of systemic acquired resistance inducing chemicals, potato leaves of different treatments were analysed for the biochemical changes that might have occured in the host tissues. The present study revealed that all the tested chemicals viz., SA, JA, BABA, GABA and H2O2 induced systemic resistance in potato plants by enhanced accumulation of different components and reduced disease incidence as compared to pathogen treated (R. solani) control and uninoculated (without R. solani) control (with water).

Total phenols

The results presented in Table 2 revealed that SAR inducing chemicals applied at both the concentrations resulted in higher content of total phenols. Among tuber dipping treatments, maximum level of total phenols was recorded with jasmonic acid (84.40 and 96.13 µg catachol g−1 FW) followed by salicylic acid (81.00 and 92.40 µg catachol g−1 FW) hydrogen peroxide (83.33 and 92.67 µg catachol g−1 fresh weight), BABA (69.27 and 87.00 µg catachol g−1 FW) and GABA (58.40 and 79.60 µg catachol g−1 FW) as compared to control at both the concentrations i.e., 150 and 250 µg/ml, respectively. Total phenolic content was substantially elevated in the inoculated control (59.07 µg catachol g−1 FW) as compared to uninoculated control (50.93 µg catachol g−1 FW) after dipping in water. However, it was shown that following the peak concentration, the total phenol content decreased in all of the treatments. It was noted that after 45 days of planting, the maximum amount of total phenols was reached in all treatments (Table 2). Similar trends were seen for the tuber dipping and foliar spray treatments, as indicated in Table 3. The total phenolic content rose in all treatments up to the third day after spraying and then fell.

Table 2 Effect of systemic acquired resistance inducing chemicals on total phenol content in potato leaves.

S. no.	Treatment	Concentration (µg/ml)	Total phenol (µg catechol g−1 FW)a	
			Days after tuber dipping	Days after foliar spray	
			30	45	60	Mean	0	1	3	6	Mean	
1	Salicylic Acid	150	67.80	97.60	77.60	81.00	88.60	112.20	147.20	119.80	116.95	
250	80.80	111.00	85.40	92.40	101.20	141.80	165.60	134.40	135.75	
2	Jasmonic Acid	150	73.20	100.80	79.20	84.40	90.40	115.60	153.00	123.20	120.55	
250	84.60	119.20	98.60	100.80	99.80	147.20	169.80	140.60	139.35	
3	BABA	150	63.80	75.80	68.20	69.27	83.60	103.60	135.40	98.60	105.30	
250	75.20	107.60	78.20	87.00	93.60	132.60	148.40	135.60	127.55	
4	GABA	150	54.60	63.20	57.40	58.40	80.60	87.20	131.00	111.60	102.60	
250	69.60	96.20	73.00	79.60	89.00	107.80	142.20	113.40	113.10	
5	H2O2	150	74.20	99.60	76.20	83.33	86.40	109.60	148.80	111.40	114.05	
250	82.80	113.80	81.40	92.67	99.20	131.20	164.80	131.60	131.70	
6	Control (Inoculated)	47.80	70.20	59.20	59.07	66.80	90.80	117.60	107.20	95.60	
7	Control (Un-inoculated)	42.20	61.40	49.20	50.93	57.40	69.00	91.20	79.60	74.30	
	Days (A)	Treatments (B)	Conc. (C)	Interaction (A × B)	Interaction (A × C)	Interaction (B × C)	Interaction (A × B × C)	
	T.D	F.S	T.D	F.S	T.D	F.S	T.D	F.S	T.D	F.S	T.D	F.S	T.D	F.S	
SE (m)	0.75	0.91	1.06	1.11	0.61	0.64	1.84	2.22	1.06	1.28	1.50	1.57	2.60	3.15	
CD (p = 0.05)	2.12	2.55	2.99	3.12	1.87	1.98	5.19	6.24	NS	3.61	4.24	4.42	7.34	8.83	
Notes.

a All values represent means of three replications.

NS Non-significant

T.D Tuber dipping

F.S Foliar spray

Table 3 Effect of systemic acquired resistance inducing chemicals on peroxidase activity in potato leaves.

S. no.	Treatment	Concentration (µg/ml)	Peroxidase activity (Change in absorbance min−1 g−1 FW)a	
			Days after tuber dipping	Days after foliar spray	
			30	45	60	Mean	0	1	3	6	Mean	
1	Salicylic Acid	150	0.282	0.438	0.378	0.366	0.372	0.545	0.719	0.598	0.559	
250	0.312	0.473	0.412	0.399	0.384	0.586	0.738	0.692	0.600	
2	Jasmonic Acid	150	0.279	0.456	0.388	0.374	0.378	0.572	0.723	0.584	0.564	
250	0.329	0.489	0.434	0.417	0.397	0.608	0.748	0.676	0.607	
3	BABA	150	0.196	0.278	0.258	0.244	0.288	0.381	0.459	0.327	0.364	
250	0.223	0.373	0.341	0.312	0.331	0.522	0.648	0.425	0.482	
4	GABA	150	0.188	0.266	0.263	0.239	0.248	0.284	0.349	0.264	0.286	
250	0.212	0.318	0.294	0.275	0.327	0.467	0.498	0.479	0.443	
5	H2O2	150	0.236	0.381	0.349	0.322	0.361	0.487	0.678	0.530	0.514	
250	0.285	0.481	0.427	0.398	0.380	0.591	0.748	0.648	0.592	
6	Control (Inoculated)	0.198	0.259	0.229	0.229	0.221	0.325	0.441	0.304	0.323	
7	Control (Un-inoculated)	0.178	0.228	0.209	0.205	0.213	0.290	0.391	0.323	0.304	
	Days (A)	Treatments (B)	Conc. (C)	Interaction (A × B)	Interaction (A × C)	Interaction (B × C)	Interaction (A × B × C)	
	T.D	F.S	T.D	F.S	T.D	F.S	T.D	F.S	T.D	F.S	T.D	F.S	T.D	F.S	
SE (m)	0.002	0.003	0.003	0.004	0.002	0.002	0.006	0.008	0.003	0.004	0.005	0.005	0.008	0.011	
CD (p = 0.05)	0.007	0.009	0.009	0.011	0.005	0.006	0.016	0.021	NS	0.012	0.013	0.015	0.023	0.030	
Notes.

a All values represent means of three replications.

NS Non-significant

T.D Tuber dipping

F.S Foliar spray

Peroxidase activity

Evaluation of effect of different levels of SAR inducing chemicals on POD activity revealed increased activity in all treatments at both the concentrations (150 and 250 µg/ml) of SA, JA, BABA, GABA and H2O2 andsuperiority over control (Table 3). Among all the tuber dipping treatments, jasmonic acid resulted maximum POD activity (0.374 and 0.417 change in absorbance min−1 g−1FW), followed by salicylic acid (0.366 and 0.399 change in absorbance min−1 g−1FW), H2O2(0.322 and 0.398 change in absorbance min−1 g−1FW), BABA (0.244 and 0.312 change in absorbance min−1 g−1FW) and GABA (0.239 and 0.275 change in absorbance min−1 g−1FW) as compared to control at both the concentrations i.e., 150 and 250 µg/ml, respectively. In the current study, it was found that after dipping potato tubers, infected controls had increased POD activity (0.229 change in absorbance min-1 g-1 FW) compared to uninoculated controls (0.0205 change in absorbance min−1 g−1 FW). All of the SAR activators were discovered to significantly enhance the POD activity compared to the respective controls. Further observations revealed that POD activity peaked in all treatments after 45 days of planting and that, after the highest concentration, all treatments had a decline in activity. The foliar spray treatment showed a similar tendency. Furthermore, it was shown that in all of the treatments, POD activity rose up to the third day after spray and then fell off.

Polyphenol oxidase activity

The effect of different levels of SAR inducing chemicals on polyphenol oxidase activity was studied. An increase in PPO activity was observed in all treatments of SA, JA, BABA, GABA and H2O2at both the concentrations (150 and 250 µg/ml) which showed superiority over to control (Table 4). Among all the tuber dipping treatments, jasmonic acid after inoculation with R. solani revealed maximum PPO activity  (0.373 and 0.402 change in absorbance min−1 g−1FW), followed by salicylic acid (0.365 and 0.396 change in absorbance min−1 g−1FW), hydrogen peroxide (0.321 and 0.377 change in absorbance min−1 g−1FW), BABA (0.293 and 0.344 change in absorbance min−1 g−1FW) and GABA (0.265 and 0.312 change in absorbance min−1 g−1FW) as compared to control at both the concentrations i.e., 150 and 250 µg/ml, respectively. The present study also revealed increased PPO activity of inoculated control (0.248 change in absorbance min−1 g−1FW) as compared to un-inoculated control (0.225 change in absorbance min−1 g−1FW) of potato leaves after dipping in water. All the SAR activators were found significant in enhancing PPO activity over both the controls. Among all SAR inducing chemicals, minimum level of PPO activity was observed in GABA followed by BABA.

Table 4 Effect of systemic acquired resistance inducing chemicals on polyphenol oxidase activity in potato leaves.

S. no.	Treatment	Concentration (µg/ml)	Polyphenol oxidase activity (Change in absorbance min−1 g−1 FW)a	
			Days after tuber dipping	Days after foliar spray	
			30	45	60	Mean	0	1	3	6	Mean	
1	Salicylic Acid	150	0.299	0.418	0.378	0.365	0.409	0.523	0.715	0.545	0.548	
250	0.319	0.451	0.419	0.396	0.447	0.644	0.782	0.623	0.624	
2	Jasmonic Acid	150	0.297	0.427	0.394	0.373	0.415	0.561	0.739	0.576	0.573	
250	0.312	0.457	0.438	0.402	0.456	0.630	0.797	0.657	0.635	
3	BABA	150	0.258	0.313	0.307	0.293	0.354	0.408	0.544	0.412	0.430	
250	0.294	0.396	0.342	0.344	0.397	0.486	0.602	0.459	0.486	
4	GABA	150	0.218	0.308	0.268	0.265	0.329	0.411	0.501	0.418	0.415	
250	0.274	0.359	0.303	0.312	0.383	0.483	0.578	0.437	0.470	
5	H2O2	150	0.261	0.387	0.316	0.321	0.379	0.498	0.698	0.516	0.523	
250	0.309	0.436	0.386	0.377	0.423	0.553	0.748	0.586	0.578	
6	Control (Inoculated)	0.204	0.293	0.248	0.248	0.314	0.358	0.467	0.450	0.397	
7	Control (Un-inoculated)	0.184	0.259	0.233	0.225	0.294	0.325	0.419	0.347	0.346	
	Days (A)	Treatments (B)	Conc. (C)	Interaction (A × B)	Interaction (A × C)	Interaction (B × C)	Interaction (A × B × C)	
	T.D	F.S	T.D	F.S	T.D	F.S	T.D	F.S	T.D	F.S	T.D	F.S	T.D	F.S	
SE (m)	0.002	0.005	0.002	0.006	0.001	0.003	0.004	0.012	0.002	0.007	0.003	0.009	0.006	0.017	
CD (p = 0.05)	0.005	0.014	0.007	0.017	0.004	0.010	0.012	0.034	NS	NS	0.010	0.024	0.017	0.048	
Notes.

a All values represent means of three replications.

NS Non-significant

T.D Tuber dipping

F.S Foliar spray

Phenylalanine ammonia lyase activity

The effect of different levels of SAR inducing chemicals was examined on PAL activity. Results presented in Table 5 showed an increase in PAL in all treatments at both the concentrations (150 and 250 µg/ml) of SA, JA, BABA, GABA and H2O2which revealed superiority over control. Among all the tuber dipping treatments, jasmonic acid after inoculation with R. solani revealed maximum PAL activity (0.392 and 0.457 change in absorbance min−1 g−1FW), followed by salicylic acid (0.388 and 0.433 change in absorbance min−1 g−1FW), hydrogen peroxide (0.349 and 0.421 change in absorbance min−1 g−1FW), BABA (0.285 and 0.339 change in absorbance min−1 g−1 FW) and GABA (0.268 and 0.319 change in absorbance min−1 g−1FW) as compared to both of the control at both the concentrations i.e., 150 and 250 µg/ml, respectively. The PAL activity was more in inoculated control (0.258 change in absorbance min−1 g−1FW) against black scurf of potato as compared to un-inoculated control (0.209 change in absorbance min−1 g−1FW) of potato leaves when dip in water. All the SAR activators enhanced PAL activity significantly over both the controls. Among all SAR inducing chemicals, minimum level of PAL activity was observed in GABA, followed by BABA.

Table 5 Effect of systemic acquired resistance inducing chemicals on phenylalanine ammonia-lyase activity in potato leaves.

S. no.	Treatment	Concentration (µg/ml)	Phenyl alanine ammonia lysase activity (µmol trans-cinnamic acid min−1 g−1 FW)a	
			Days after tuber dipping	Days after foliar spray	
			30	45	60	Mean	0	1	3	6	Mean	
1	Salicylic Acid	150	0.289	0.483	0.392	0.388	0.395	0. 695	0.837	0.694	0.642	
250	0.329	0.532	0.438	0.433	0.424	0.730	0.876	0.752	0.696	
2	Jasmonic Acid	150	0.301	0.496	0.378	0.392	0.407	0.664	0.691	0.666	0.607	
250	0.338	0.553	0.481	0.457	0.433	0.738	0.849	0.683	0.676	
3	BABA	150	0.227	0.329	0.299	0.285	0.344	0.448	0.535	0.473	0.450	
250	0.269	0.389	0.358	0.339	0.387	0.524	0.593	0.658	0.541	
4	GABA	150	0.226	0.316	0.263	0.268	0.334	0.425	0.483	0.538	0.445	
250	0.263	0.387	0.307	0.319	0.382	0.477	0.558	0.647	0.516	
5	H2O2	150	0.269	0.409	0.368	0.349	0.353	0.539	0.635	0.552	0.520	
250	0.317	0.504	0.443	0.421	0.397	0.599	0.687	0.738	0.605	
6	Control (Inoculated)	0.223	0.298	0.253	0.258	0.284	0.374	0.469	0.478	0.401	
7	Control (Un-inoculated)	0.164	0.249	0.213	0.209	0.257	0.308	0.385	0.536	0.372	
	Days (A)	Treatments (B)	Conc. (C)	Interaction (A × B)	Interaction (A × C)	Interaction (B × C)	Interaction (A × B × C)	
	T.D	F.S	T.D	F.S	T.D	F.S	T.D	F.S	T.D	F.S	T.D	F.S	T.D	F.S	
SE (m)	0.002	0.004	0.002	0.005	0.001	0.003	0.004	0.009	0.002	0.005	0.003	0.007	0.006	0.013	
CD (p = 0.05)	0.005	0.011	0.006	0.013	0.004	0.008	0.011	0.026	0.006	0.015	0.009	0.019	0.016	0.037	
Notes.

a All values represent means of three replications.

T.D Tuber dipping

F.S Foliar spray

Discussion

In the current study, SAR activators salicylic acid and jasmonate along with hydrogen peroxide were the most efficient in reducing the incidence of disease, while GABA and BABA were the least effective. Important signal molecules like SA, jasmonate, and ethylene, which all significantly alter gene expression and are involved in complicated crosstalk, control the pathways in induced resistance mechanisms (Godoy et al., 2000; Ismail & Hijri, 2012; Rehaman et al., 2021). It is linked to the beginning of a group of genes called pathogenesis-related (PR) genes, which include chitinases, acidic and basic-1-3-glucanases, and a large number of additional genes with unclear functions (Ismail & Hijri, 2012).

The results revealed minimum disease incidence with SA (26.67 and 16.67%), followed by JA (30.00 and 21.67%), H2O2(31.67 and 25.00%), BABA (38.34 and 33.34%) and GABA (41.67 and 35.00%) as compared to inoculated control (63.33%) and un-inoculated control (6.70%) at both the concentrations i.e., 150 and 250 µg/ml, respectively. All the SAR activators were significantly superior in reducing disease incidence. The highest reduction in disease incidence was observed with SA, followed by JA, H2O2, and BABA compared to the control at both concentrations. Lowest reduction in disease incidence was observed with GABA (34.20 and 44.73%). The results are consistent with previous research, which examined how well SA worked to lessen the severity of black scurf in a screen house environment (Kiptoo et al., 2021). Treatments with salicylic acid greatly reduced the severity of R. solani in both tuber dipping and foliar spray.

The results indicated that SAR activators such as SA, JA, H2O2, BABA, and GABA showed varying degrees of effectiveness in reducing disease incidence. SA consistently exhibited the highest reduction, followed by JA and H2O2, across both concentrations tested (150 and 250 µg/mL). Conversely, GABA and BABA demonstrated comparatively lower efficacy. Overall, all SAR activators were significantly superior to the inoculated control in reducing disease incidence

Previously, Matny & Al-Jarrh (2014) discovered that treating scurf infected potato tubers with H2O2 at concentrations of 150 and 250 µg/mL, reduced disease incidence. They found that the therapies using SA and H2O2 were more effective at reducing the severity of the condition than the treatments using BABA. Additionally, it was determined that SA, H2O2, and BABA had a substantial impact on raising the peroxidase enzyme activity in potato leaf. The new findings also show that H2O2 plays a part in triggering a number of host defensive mechanisms, including the activation of enzymes like peroxidase and a marked increase in lignin and suberin levels (Li et al., 2010).

It is well recognised that different SAR activators, including aminobutyric acid, salicylic acid, and peroxide hydrogen, cause systemic resistance in a variety of plant species (Tiwari et al., 2021b). To reduce Fusarium oxysporum f. sp. lycopersici in tomatoes, foliar spray with SA caused increased PAL and POD activity that were 3.7 and 3.3 times greater than with control treatment (Mandal, Mallick & Mitra, 2009). After 72 h of R. solani inoculation, the application of acetyl salicylic acid, amino isobutyric acid, IAA, BABA, and SA led to lesion-free leaves. Maximum disease control was achieved with seed treatment and foliar spraying with gamma-aminobutyric acid, -butyric acid, and isonicotinic acid, respectively (Dantre, Rathi & Sinha, 2003). El-Mohamedy, Jabnoun-Khiareddine & Daami-Remadi (2014) examined SAR-inducing substances against R. solani, which causes tomato root rot, and discovered that potassium salts, salicylic acid, and sorbic acid inhibited the pathogen responsible for the disease. The incidence and severity of root rot under greenhouse conditions were significantly reduced by all chemical inducers.

Tuber dipping of SAR activators was found to be most effective in managing disease as compared to foliar spray. While it was observed that there was more increase in the content of all biochemical compositions in foliar spray as compared to tuber dipping with all the SAR-inducing chemicals treatments. In general, plants require time to trigger disease resistance when it responds to pathogen infection. It might be due to SAR activators induce a defense mechanism in initial stage of plant growth against inoculated pathogen and its persist up to tuber maturation in tuber dipping treatment, whereas, in case of foliar spray SAR activators induce a defense mechanism at tuber initiation stage of plant growth against inoculated pathogen till to that the pathogen is established. While in the foliar spray at the tuber initiation stage, the pathogen was already established, and due to biotic stress, biochemical activity increased quickly by SAR activators and enhanced its defence mechanism to combat the pathogen. Yu et al. (2017) also held the same presumptions. It’s interesting to note that these results were also in line with a previous study by Sood, Sohal & Lore (2013), who described the effects of applying benzothiadiazole (BTH) and SA to rice plants’ leaves to prevent the sheath blight disease caused by R. solani. They proposed using BTH and SA to pre-treat rice leaves to increase the amount of protection against sheath blight.

The present study revealed that all the tested chemical inducer viz., SA, JA, BABA, GABA and H2O2 were found effective to induce systemic resistance by enhanced different biochemical compositions and reduced disease incidence as compared to inoculated (with R. solani) and uninoculated (without R. solani) control (with water). The present study revealed that all chemical inducers applied at 250 µg/ml concentration resulted in higher content of all biochemical compositions after treatments which were significant as compared to 150 µg/ml concentration of all chemical inducers. It was also observed that there was more increase in all biochemical compositions content in foliar spray as compared to tuber dipping in all the SAR inducing chemical treatments. The present findings are in corroboration with the results reported by El-Naggar et al. (2013) who revealed that the resistant Draga cultivar of potato against R. solani showed the highest concentrations of total phenols, oxidative enzymes (peroxidase and polyphenol oxidase) compared with the highly susceptible lady Rosetta.

An increase in total phenol content with SAR inducing chemicals at both the concentrations (150 µg/ml and 250 µg/ml) of SA, JA, BABA, GABA and H2O2found superior over to the control treatment. The highest level of total phenol was recorded with JA(84.40 and 96.13 µg catachol g−1FW), followed by SA(81.00 and 92.40 µg catachol g−1FW), H2O2(83.33 and 92.67 µg catachol g−1FW), BABA (69.27 and 87.00 µg catachol g−1FW) and GABA (58.40 and 79.60 µg catachol g−1FW) as compared to control at both the concentrations i.e., 150 and 250 µg/ml, respectively, with tuber dipped treatments. All the SAR activators significantly enhanced total phenol content over both the controls. Total phenols content recorded was increased significantly (P < 0.005) with time and highest concentration of total phenol content in all the treatments was observed at 45 days of planting and after that it decreased. Total phenol content was significantly more in the inoculated control (59.07 µg catachol g−1 FW) as compared to uninoculated control (50.93 µg catachol g−1 FW) after dipping in water. The outcomes are consistent with those who found that inoculated plants had higher levels of total phenol content in their leaves and roots than healthy, non-inoculated plants (Bagy et al., 2019). According to Demidchik (2012), total phenol content increased significantly (P < 0.005) over the course of all treatments, reaching its peak at the third day after spray and subsequently declining. Phenolic compounds also function as antioxidants by scavenging reactive oxygen species. Activity of defense related enzyme peroxidase was observed after the treatment with the SAR inducing chemicals. Among all the tuber dipping treatments, JA revealed highest POD activity (0.374 and 0.417 change in absorbance min−1 g−1FW), followed by SA (0.366 and 0.399 change in absorbance min−1 g−1FW), while H2O2(0.322 and 0.398 change in absorbance min−1 g−1FW) BABA (0.244 and 0.312 change in absorbance min−1 g−1FW) and GABA (0.239 and 0.275 change in absorbance min−1 g−1FW) exhibited lower activity at both the concentrations. The activity of POD increased significantly (P < 0.005) with time and peak concentration of POD activity in all the treatments was observed at 45 days of planting and after that it decreased. It was also noted that foliar spray treatment led to substantial changes in the POD activity. A similar type of trend was recorded for foliar spray treatment to tuber dipping treatment. It was observed that in all the treatments, the POD activity increased significantly (P < 0.005) with time and peak activity recorded at 3rd days after spray and then it decreased.

Results of the present study showed an increase in PPO activity in all the treatments at both the concentrations (150 and 250 µg/ml) of salicylic acid, jasmonic acid, BABA, GABA and hydrogen peroxide, which showed better results over the control treatment. All the SAR activators were found significant in enhancing PPO activity over both the controls. Among all the tuber dipping treatments, JA after inoculation with R. solani revealed maximum PPO activity(0.373 and 0.402 change in absorbance min−1 g−1FW), followed by SA (0.365 and 0.396 change in absorbance min−1 g−1FW), while H2O2(0.321 and 0.377 change in absorbance min−1 g−1FW), BABA (0.293 and 0.344 change in absorbance min−1 g−1FW) and GABA (0.265 and 0.312 change in absorbance min−1 g−1FW) showed lower activity as compared to control at both the concentrations. The PPO enzyme activity reached the maximum at 45 days after planting and it remained at higher level up to 60 days after challenge inoculation with R. solani and decreased thereafter with all the SAR inducing chemicals.

It was revealed that all the tested SAR inducer chemicals were effective to induce the activity of PAL enzyme. Among all the tuber dipping treatments, JA after inoculation with R. solani revealed maximum PAL activity (0.392 and 0.457 change in absorbance min−1 g−1FW), followed by SA (0.388 and 0.433 change in absorbance min−1 g−1FW), while lower activity of H2O2(0.349 and 0.421 change in absorbance min−1 g−1FW), BABA (0.285 and 0.339 change in absorbance min−1 g−1 FW) and GABA (0.268 and 0.319 change in absorbance min−1 g−1FW) was noticed as compared to both of the control at both the concentrations (150 and 250 µg/ml). The activity of PAL enzyme in SAR inducing chemicals treatments significantly (P < 0.005) increased continuously in the initial time of inoculation and reached a peak value of enzyme concentrations at 45 days of planting. There was a decline in PAL activity in all the treatments after the peak concentration. The outcomes are consistent with those of Ketabchi, Majzoob & Charegani (2015) who investigated the impact of salicylic acid and methyl jasmonate on total phenol and phenylalanine ammonia lyase activity in wheat infected by Pratylenchus thornei. Two days following inoculation, they noticed that all treatments had considerably higher levels of PAL and total phenol activity. However, five days following the injection, PAL activity was lower in all treatments when compared to controls. Though SA and JA pathways are often associated with different types of stressors (in general, SA for biotrophs and JA for necrotrophs), frequent crosstalk between these pathways allows plants to modulate their defence responses. After treatment with SAR-inducing chemicals, both pathways can be activated at the same time to provide a comprehensive defense strategy.

Figure 2 An illustrative model of systemic acquired resistance inducing chemicals in mitigation of black scurf disease and induction of defence network.

The spray, as well as dip treatment, has been shown to activate the defence-related enzymes viz. PPO, PAL, and POD ultimately lead to a resistant phenotype. Abbreviations: BABA, β-Aminobutyric acid; GABA, γ-Aminobutyric acid and H2O2, Hydrogen peroxide; PPO, Polyphenol oxidase; PAL, Phenylalanine ammonia-lyase; POD, Peroxidase.

Conclusion

In the current scenario of sustainable disease management, the role of plant defence activators and SAR chemicals is immense in the alleviation of pathogenic infections. These activators not only enhance plant growth and mitigate the disease but also help in minimizing the pesticide burden on the microenvironment. In this context, the present study highlighted the active role of salicylic acid, jasmonic acid, β-aminobutyric acid, γ-aminobutyric acid and hydrogen peroxide in suppression of a critical pathogen of potato which cause qualitative and quantitative loss to the crop. The tuber dip and foliar spray of these SAR-inducing chemicals were effective in the reduction of disease incidence via the activation of defense-related enzymes. The significant differences were evident in tuber dip and foliar spray application in a concentration-dependent manner. The results of disease alleviation were correlated with enhanced peroxidase, polyphenol oxidase and phenyl ammonia lyase enzymes which are active components of induced defense responses in infected plants(Fig. 2). A sustainable integrated disease management program requires these chemicals for effective environment-friendly management of these pathogens.

Supplemental Information

Data S1 Raw data of analysis of disease and enzyme related parameters

Authors acknowledge the director of ICAR-CPRI Shimla for access to literary resources.

Additional Information and Declarations

Competing Interests

Author Contributions

Data Availability

Ravinder Kumar is an Academic Editor for PeerJ.

Manoj Kumar Buswal conceived and designed the experiments, performed the experiments, prepared figures and/or tables, authored or reviewed drafts of the article, and approved the final draft.

Rakesh Punia conceived and designed the experiments, authored or reviewed drafts of the article, and approved the final draft.

Mukesh Kumar performed the experiments, prepared figures and/or tables, and approved the final draft.

Rahul Kumar Tiwari analyzed the data, prepared figures and/or tables, and approved the final draft.

Milan Kumar Lal analyzed the data, authored or reviewed drafts of the article, and approved the final draft.

Ravinder Kumar conceived and designed the experiments, analyzed the data, prepared figures and/or tables, authored or reviewed drafts of the article, and approved the final draft.

The following information was supplied regarding data availability:

The raw data is available in the Supplemental File.

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
