# Peer review of "Systemic acquired resistance inducing chemicals mitigate black scurf disease in potato by activating defense-related enzymes"

_PeerJ, doi:10.7717/peerj.18470_

## Round 0.1 · original submission · Major Revisions

· Academic Editor

Major Revisions

The reviewers and I think that this paper is a preliminary research work and there are many necessary contents that need to be supplemented. Please revise the author based on the reviewer's opinions, and add experiments to improve the research quality of the paper.

**Language Note:** The review process has identified that the English language must be improved. PeerJ can provide language editing services - please contact us at [email protected] for pricing (be sure to provide your manuscript number and title). Alternatively, you should make your own arrangements to improve the language quality and provide details in your response letter. – PeerJ Staff

Reviewer 1 ·

Basic reporting

In this manuscript, Buswal et al. explore the effectiveness of systemic acquired resistance-inducing chemicals, including salicylic acid, jasmonic acid, betaThe -aminobutyric acid, gamma-aminobutyric acid, and hydrogen peroxide, in mitigating black scurf and stem canker disease in potatoes caused by Rhizoctonia solani. They found that these chemicals significantly reduce disease incidence in a concentration-dependent manner, with salicylic acid being the most effective, particularly when applied through tuber dipping, which was more efficacious than foliar spray, supporting their inclusion in integrated disease management programs. The topic of this manuscript aligns well with the interests of PeerJ readers. However, revisions are necessary to get this work published. Please find my comments below.

There are a lot of grammar errors. The English language should be improved to ensure that an international audience can clearly understand your text. Examples of areas needing improvement include lines 18, 47, 59, 303 and 356, where the current phrasing is hard to understand. I recommend having a fluent English speaker review your manuscript or engaging a professional editing service.

The figure captions require more detailed explanations to adequately describe the figures.

The authors need to include visualizations of the raw data presented in the tables through figures for clearer interpretation.

Experimental design

None

Validity of the findings

The discussion section predominantly reports results already documented by other researchers, as acknowledged in the manuscript. The novelty and impact of the new findings from this study must be more thoroughly articulated to justify a standalone publication in PeerJ

Additional comments

None

Reviewer 2 ·

Basic reporting

The English used throughout the manuscript could be improved. Then it might be easier to read and the description of the methods could perhaps be followed. Literature references are sufficient, the structure is ok, but the tables are not readable for a reviewer due to the formating

Experimental design

The research question is well defined. However,the investigation has not been performed as expected.
1. the methods are not presented in a way that one can follow the procedures and calculations
2. highly important information of the methods and experimental procedures are missing
2.1 how was the inocculum made and applied?
2.2 what are the used solvents to apply the different chemicals?
2.3 uninocculated but chemical treated controls are missing

Validity of the findings

Since the methods and procedures are lacking clear description, the validity of findings cannot be reviewed. Control treatments ( see 2.2.2 and 2.2.3) are missing.
The problem of growth-defense trade-off is not mentioned or discussed. Yield differences are to be expected in plants that invest much more in defense. The strong induction of phenolic compound synthesis and POX, PPO and PAL activity in all SAR-treatments is much higher than the induced inductions upon pathogen inocculation. For a clear result, yield measurements and comparisons with the missing control treatments are necessary.

Additional comments

minor points are typing errors (line 85 thiophanate, line 204 cc) or deficits with citations (line 80, 93, 94, 325) where the year is missing or a clear statement at what time the online reference has been accessed.

Reviewer 3 ·

Basic reporting

1. Many sentences are difficult to understand. The language of this manuscript should be further polished.
2. Some conclusions need raw data to support, but the authors don't provide it.

Experimental design

No comment.

Validity of the findings

We can’t get three repeated results from submitted files when they detected different parameters (Only three or four days obtained an average value, but we can't understand why they did like this). It’s illogical for scientific research. Only one replication may be an occasional result. The authors should provide these original results.

Additional comments

Major questions:
1. The authors used 5 inducing chemicals to evaluate the disease incidence after treated by R. solani. How about the phenotypes of potato tuber after treatment? The authors should show this data, because phenotype is direct evidence to show the efficiency of inducing resistance.

2. The authors also should provide the data of disease incidence with a table format, otherwise, we can’t get the conclusion of line 29-33.

3. Line33-34, They mentioned that “The tuber dipping treatments were significantly higher (P<0.005) efficacious as compared to foliar spray for all treatment combinations.” How they verify that method is crucial not caused by the long-time treatment? Because tuber dipping treatments were at least 30 days, while foliar spray treatment just 6 days.

4. The data investigated that the concentration of 150 and 250 µg/mL were all inducing systemic resistance in potato than control, and 250 µg/mL was better than 150 µg/mL. So whether it means that higher concentration of these inducing chemicals were more favorable to enhance the disease resistance to pathogen in potato? I think the authors should add a higher concentration (500 µg/mL?) to verify the best range of concentration during application.

5. Rhizoctonia solani is a necrotrophic pathogen, why SA and JA all induced resistance in potato, commonly, JA is crucial for the resistance of plants to necrotrophic pathogens. After inoculated by Rhizoctonia solani, whether SA/JA-responsive genes increased under these five treatments? They also don’t mention it.


Minor questions:
1. The language of this manuscript should be further polished.

2. Whether these 5 inducing chemicals have a direct fungistasis to Rhizoctonia solani?

3. Fig.1, it missed the notes of “ns”, “*” and the statistical approach.

4. Writing format, Line32, 181 184 185. Please check it in the whole manuscript.

5. Reference format, Line79, 91, 93, 321, and 382. Please check it in the whole manuscript.

6. Line240 and 300, the format of text (bold font) is not consistent with other subtitles.

7. Some abbreviations have appeared at background part, but the full name and abbreviation name also appeared at result and discussion part, for example, line247, 248, 263, 275, 387, 388, etc. Please revise it in the whole manuscript.

---

## Round 0.2 · Minor Revisions

· Academic Editor

Minor Revisions

The reviewers are largely satisfied with your revisions, but there are still some minor issues that need to be revised, so please follow the suggestions.

Reviewer 1 ·

Basic reporting

The authors did a great job on the revision. The current form of the manuscript is acceptable for publication with minor revisions, which I will describe below:

In Figure 1, please add the abbreviations for BABA, GABA, etc., as these terms are appearing in figures for the first time.
In Figure 2, use "An illustrative" instead of "a illustrative." Also, address the abbreviation issues in this figure as well.

Experimental design

no comment

Validity of the findings

no comment

Additional comments

no comment

Reviewer 2 ·

Basic reporting

As I am not sure how the experiments were carried out, I cannot really judge the results obtained. The raw data are missing, only mean values are shown for which no standard deviation is given. It is not clear whether the mean value consists of only three measurements

Experimental design

The original primary research fits to aims and scopes. The question is well defined. The performed investigation and the description of the methods is insufficiant. There is still no clear description of how the experiments were carried out. My interpretation of how the experiments were conducted and the data collected raises a severe problem with these data. The time 0 of foliar spraying should match one of the untreated samples. But at time 0 of foliar spraying, all values are already much higher than in the control.

Validity of the findings

Due to the shortcomings described in point 2, I am not in a position to validate the data and the conclusions drawn.

Reviewer 3 ·

Basic reporting

Comments to the Author
The authors have addressed all my concerns, and it is acceptable for publication.

A few minor suggestions:

1. The unit should be used the same format. For example, the unit of “mL" in line 183 and line 185 with M is capital letter, and others are not. And the unit of min-1 g-1 at line 277-278 is different from line 272-273 (min-1 g-1). Please revise these questions in the whole manuscript.

2. At the section of 2.4 Statistical analysis. Please add the statistical approach: what’s the meaning of the “*” and “***”. In your Figure 1, you don’t mention what’s the “*” and “***” represent.

3. At the section of Discussion. Expand the discussion to analyze the significance of the results in greater depth, such as why the salicylic acid (SA) and Jasmonic acid (JA) signaling pathway are all activated after treatments.

Experimental design

no comment

Validity of the findings

no comment

Additional comments

no comment

---

## Round 0.3 · accepted · Accept

· Academic Editor

Accept

All three reviewers and I felt that your article met the requirements for acceptance, congratulations!

Reviewer 1 ·

Basic reporting

The authors addressed all my concerns and I recommend the publication of this manuscript.

Experimental design

no comment

Validity of the findings

no comment

Reviewer 2 ·

Basic reporting

ok

Experimental design

ok, but an extended, improved description of the experiments could still be helpful

Validity of the findings

ok

Reviewer 3 ·

Basic reporting

The authors have addressed all my concerns, and it is acceptable for publication.

Experimental design

No comment

Validity of the findings

No comment

Additional comments

No comment